# Composite Polymeric Cryogel Cartridges for Selective Removal of Cadmium Ions from Aqueous Solutions

**DOI:** 10.3390/polym12051149

**Published:** 2020-05-18

**Authors:** Sabina Huseynli, Monireh Bakhshpour, Tahira Qureshi, Muge Andac, Adil Denizli

**Affiliations:** 1Department of Chemistry, Biochemistry Division, Hacettepe University, 06800 Ankara, Turkey; sabinah@hacettepe.edu.tr (S.H.); b.monir@hacettepe.edu.tr (M.B.); qureshitahira@gmail.com (T.Q.); 2Department of Environmental Engineering, Hacettepe University, 06800 Ankara, Turkey; andac@hacettepe.edu.tr

**Keywords:** Ion imprinting, Composite cryogel cartridge, Cd(II) ions, Selective adsorption, Wastewater

## Abstract

In this study, composite polymeric cryogel cartridges were achieved by using Cd(II) imprinted poly(hydroxyethyl methacrylate N-methacryloly-(L)-cysteine methylester) beads and poly(hydroxyethyl methacrylate) cryogel cartridges with two different mole ratios of functional monomer. The N-methacryloly-(L)-cysteinemethylester was used as a functional monomer and Cd(II) 1:1 and 2:1, which were then notated as MIP1 and MIP2, respectively. Various characterization methods have confirmed the structural transformation on the MIP1 and MIP2 composite cryogel cartridges by scanning electron microscopy, Fourier-transform infrared spectroscopy-Attenuated Total Reflectance, and swelling tests. The maximum amount of Cd(II) adsorption with composite cryogel cartridges was determined by altering the Cd(II) initial concentration, temperature, and pH values. The maximum adsorption capacity of MIP1 and MIP2 composite cryogel cartridges obtained was 76.35 and 98.8 µmol/g of composite cryogels, respectively. The adsorption studies revealed that the MIP2 possessed a good adsorption performance for Cd(II). The obtained composite cryogel cartridges have a selective, reusable, and cost-friendly potential for the removal of Cd(II) from aqueous solutions, and are used many times without decreasing their adsorption capacities significantly. The Cd(II) removal rate of the MIP1 and MIP2 composite cryogel cartridges from synthetic wastewater samples was determined as 98.8%. The obtained cryogel cartridges’ adsorption material exhibited a good directional removal performance for Cd(II) from wastewater samples.

## 1. Introduction

In current times, the balance of water quality is getting disturbed by various factors, such as natural calamities, anthropogenic activities, etc. These factors impact water quality negatively and damage its sustainability for all living beings [1,2]. The major contaminants found in water systems are aromatic organic compounds, pesticides, heavy metals, plastic, pharmaceutical active compounds, etc. Elevated concentrations of toxic heavy metals (Cd, Cr, Ni, Hg, and Pb) were observed in the gills and liver of some fish species as compared to their muscles [3,4,5]. Heavy metals pollution is one of the most important environmental issues due to their toxic effects and persistent nature. Toxic heavy metals easily bio-accumulate on aquatic species and lead to the spread of their toxic effect on consumptions [6]. The remediation of heavy metals requires special attention to protect the ecosystem from its contamination, to reduce its adverse impact on living beings [7]. Therefore, researchers are continuously conducting studies to find the befitting method for wastewater treatment [8]. Researchers have searched for an alternative inexpensive and effective method instead of expensive systems in treatment methods [9,10]. The removal of heavy metals from polluted wastewaters requires great efforts. The heavy metal ions-contaminated wastewater has been treated by various techniques such as ion exchange, precipitation, electrolytic, adsorption, and evaporation techniques.

Conventional methods have significant disadvantages, such as, for instance, incomplete removal, production of toxic sludge or waste products, and high-energy consumptions. Adsorption methods are widely used for the purification of industrial wastewater from metal ions [11,12,13,14,15]. According to the World Health Organization (WHO) toxic metal ions list, some of the heavy metals are a major health concern; here, cadmium (Cd) holds a prominent place. Meanwhile, the ’Food and Agriculture Organization (FAO) and the WHO have stipulated a safe intake standard, with a permissible monthly intake of cadmium ions at 25 μg.kg^−1^ of body weight [16]. The Cd enters into the ecosystem through atmospheric sources and a variety of industrial processes. Cd(II) are toxic in nature and pose a serious health threat to living beings [17,18,19]. The high amount of cadmium poisoning in humans has caused respiratory distress, hypertension, a number of cancers, cardiovascular disease, tracheobronchitis, various endocrine organs, and kidney problems [20,21,22]. As reported, severe toxicity has been rooted in Cd(II); therefore, Cd(II) removal from environmental water is a highly important matter. Therefore, the selective removal of cadmium ions in the environment has become more urgent. The conventional remediation technologies are not enough to remove Cd(II) efficiently, and hence there is a requirement to take up technology with cost effectiveness and optimum efficiency with a targeted approach for water remediation. It is obvious that remediation technologies require new, proficient, highly specific, selective, and sensitive materials for the removal of toxic and other contaminants. More particularly, inventive methods are required for the preparation of *tailor-made* functional materials that possess performance stability in the long term. Molecularly imprinted polymers (MIPs) are one of the most reliable groups of novel and extremely selective functional materials.

Molecular imprinting technology (MIT) is a general strategy to produce polymers, in order to possess specific recognition sites complementary to a target molecule. Ion imprinting technology is similar to MIT and is prepared with the same approach except using an ion as the template [23,24,25,26,27,28,29,30,31]. The ion-imprinting technique is considered to have a highly selective and specific recognition. The need for removal processes has driven the evolution of different types of chromatographic matrices. The matrix material needs to be selected so as to have a high specific surface area, as well as stable mechanical and chemical properties. Modified natural and metal oxide sorbents are cheap and characterized by a high sorption capacity towards toxic metal ions [32,33,34,35]. Cryogels have many advantages such as large pores, a short diffusion path, and a very short swelling time for the removal of metals. The porous surface of cryogel makes it efficient for sorption and creates cavities for selective attachment on the surface. On the other hand, cryogels have disadvantages such as a low surface area, and thus a low adsorption capacity for the removal of metals. To increase the adsorption capacity of cryogels, microbeads embedding would be a useful improvement mode to use in the preparation of novel composite cryogels to increase the surface area [36,37,38,39,40].

In this study, ion-imprinted polymer beads were used for the selective separation of Cd(II) ions from the solution. N-Methacryloly-(l)-cysteinemethylester (MAC) was used as the metal complexing monomer. After the removal of Cd(II) ions, ion-imprinted beads were used for the separation of Cd(II) from the solution. Cd(II) adsorption and selectivity studies of cadmium versus other metal ions are reported here. Finally, the repeated use of the ion-imprinted beads is also discussed. PHEMA cryogel cartridges were selected as the basic matrix by considering properties which make it useful for possible extracorporeal therapy, including the hydrophilic character, good blood-compatibility, minimal non-specific protein interactions, high chemical and mechanical stability, and resistance toward microbial and enzymatic attacks [39,40]. Finally, MIP1 and MIP2 composite cryogel cartridges were performed using an artificial wastewater sample (HIGH-PURITY STANDARDS, Cat. #CRM-SOIL-B) to test the selective removal efficiency of composite cryogel cartridges. Cryogels provide potential advantages in terms of their low pressure drop and lack of diffusion resistance as compared to traditional gel-bead columns. The use of cryogels is promising in cases where the selective removal of imprinted metal ions from a complex mixture like wastewater is crucial [38].

## 2. Experimental Section

### 2.1. Materials

All chemicals and solvents that were used were of analytical grade. Hydroxyethyl methacrylate (HEMA), ethylene glycol dimethacrylate (EGDMA), poly(ethylene glycol) diacrylate (PEGDA), ammonium persulfate (APS) and cadmium nitrate tetrahydrate (Cd(NO_3_)_2_.4H_2_O) were purchased from Sigma Aldrich Chemical Co. (St Louis, MO, USA). N,N,N’,N’ Tetramethylethylenediamine (TEMED) was bought from Amresco (St Louis, MO, USA). The rest of the experimental chemicals were acquired from Merck AG (Darmstadt, Germany). Deionized (DI) water was generated in real time from the Milli-Q Ultrapure water system (Millipore, Dubuque, IA, USA).

### 2.2. Preparation of the Composite Cryogel Cartridges

First, the MAC as functional monomer was synthesized according to our previous publications [41]. The functional monomer and Cd(II) ions as the template were complexed by mixing with two different molar ratios (MAC/Cd(II)) as 1:1 (0.2:0.2 mmol) and 2:1 (0.4:0.2 mmol) for 60 min. The molecular formulas of the 1MAC:1Cd(II) and 2MAC:1Cd(II) complex monomer [41] are shown in Figure 1A,B, respectively. 

PHEMAC-Cd(II) beads were prepared by suspension polymerization method [42]. We used 2.0 mL of HEMA and then added 1.0 mL of EGDMA as the monomer phase in a reaction vessel. 1.0 mL of toluene was mixed to the mixture in the reaction vessel, after which the MAC-Cd(II) complex was dissolved in this solution. Toluene and EGDMA were used as a diluent (pore former) and crosslinker, respectively. Then, APS and TEMED (1% *w*/*v*) were added into the polymer’s mixture. Then, polymerization was allowed for 24 h at room temperature. The unreacted monomer was removed with ethanol: water solution (% 50 *v*/*v*), respectively. The prepared beads were dried in a lyophilizer at 50 mbar vacuum (Christ Freeze Dryer-Alpha 1–2 LD, Maryland, USA). For the preparation of PHEMAC-Cd(II) composite cryogel cartridges (MIP), monomer (1.3 mL HEMA) and crosslinker (0.506 mL of PEGDA) were dissolved in deionized water separately. These two solutions were then mixed with 100 mg of dry PHEMAHC-Cd(II) beads with a total concentration of 6% (*w*/*w*). The polymerization was initiated with an APS: TEMED activator system and kept at −16 °C for 24 h. Both MIP1 and MIP2 composite cryogel cartridges were prepared by the same method. Additionally, the non-imprinted PHEMAC (NIP) composite cryogel cartridge was prepared by the same polymerization without complexing MAH monomer with Cd(II) ions for control experiments. As the polymerization ended, the composite cryogel cartridges were allowed to thaw and were washed with deionized water several times.

### 2.3. Characterization of MIP Composite Cryogel Cartridges

Scanning and optical electron microscopy (SEM) were examined to detect the surface morphology of PHEMAC-Cd(II) composite cryogel, notated as MIP1 and MIP2 according to their molar ratio of 1:1 and 2:1 (MAC/Cd(II)), respectively. For this purpose, both composite cryogel cartridges were desiccated in a vacuum furnace at 60 °C for 24 h. A piece of the dry composite cryogel cartridges was attached on SEM (40:60) (JEOL, JEM 1200 EX, Tokyo, Japan) and was coated with gold. After coating with altpaladium, the instances were attached in a scanning electron microscope. First, the wet composite cryogel cartridges were measured, and then they were compressed to move away from the water within the macropores. 

The structure of MIP1, MIP2, and NIP composite cryogel cartridges was studied with FTIR-ATR (FTIR 8000 Series, Shimadzu, Japan). Prior to examination, the composite cryogel cartridge was dried in a vacuum (50 mbar) in a lyophilizer for 24 h. For the FTIR-ATR, the dried polymers were pulverized on a plate. The spectra of the samples were taken at a 4000–600 cm^-1^ wave number in the FTIR-ATR device.

The water uptake ratios for the MIP1, MIP2, and NIP composite cryogel cartridges were determined by using distilled water as follows [43]: Drying is needed to weigh a composite cryogel cartridge with a sensitivity of ± 0.0001. Drying was applied at −60 °C in a lyophilizer. The amount of dried weight is calculated as W_d_. The dry composite cryogel cartridge was cautiously weighed before being put in water. After 24 h, the swollen samples were weighed again, and this weight was recorded as W_sq_. The total swelling ratio of the composite cryogel cartridges was recorded as (% SR). The equation is as follows:Swelling Ratio % = [(W_sq_ − W_d_)/W_d_] × 100(1)

### 2.4. Removal of Cd(II) Ions from Aqueous Solutions

For the removing procedure of Cd(II) ions from aqueous solutions, the effect of different concentration values between 10–200 mg/L, effect of the temperatures between 4 °C to 45 °C, and effect of the different ranges of pH values (4.0–8.0) on the adsorption rate and binding capacity were studied for a concentration of Cd(II) of 100 mg/L, T: 25 °C, and flow rate of 0.5 mL/min. The adsorption experiments were carried out with MIP1, MIP2, and NIP composite cryogel cartridges by using a peristaltic pump with a 1.0 mL/min flow rate (Watson-Marlow, Wilmington, MA, USA). The adsorption capacities of MIP1, MIP2, and NIP were calculated in Equation (2), below:(2)$$q_{e}=\frac{C_{0}-C_{e}}{m}\times V$$
where C_0_ (mg/L) and C_e_ (mg/L) are the concentrations of Cd(II) ion in the original and equilibrium solutions, V (L) is the volume of Cd(II) ion solution, and m (g) is the mass of adsorbents.

50 mM EDTA was used as the desorption solution for 1 h. The concentrations of metal ions in the aqueous solutions at the beginning of the experiment, at the end of the experiment, at the end of desorption, and at the desired time intervals were measured by flame atomic absorption spectrophotometer (AAS, Analyst 800/Perkin Elmer).

### 2.5. Selectivity Experiments 

The selectivity of MIP1 and MIP2 composite cryogel cartridges was shown with competitive adsorptions of Pb(II) with 133 pm and Zn(II) with 88 pm ionic radiuses towards Cd(II) ions with an ionic radius of 114 pm. The selection criteria of Pb(II) and Zn(II) are affinity and ionic radius. The concentration of each metal ion was kept at 100 mg/L. The binding capacity was studied in a concentration of T: 25 °C and flow rate of 0.5 mL/min. The adsorptions were also attempted by the GFAAS system. The distribution coefficients (*K_d_*) of Pb(II) and Zn(II) were shown as below (Equation (3)): 

Distribution coefficient:(3)$$K_{d}=\frac{C_{i}-C_{f}}{C_{f}}\times \frac{V}{m}\;$$
where *K_d_* represents the distribution coefficient (ml/g), C_i_ and C_*f*_ are the initial and final concentrations (mg/L), seriatim. *V* is the volume of the solution (mL) used for adsorption, and *m* is the weight of composite cryogel cartridge used (g). 

The selectivity coefficient (k) was described as below:

Selectivity coefficient:(4)$$k=\frac{Ktemplate\;metal}{K_{interferent\;metal}}$$

A relative selectivity coefficient k’ can be detected as:

Relative selectivity coefficient:(5)$$\;k\prime =\frac{k_{imprinted}}{k_{control}}$$

### 2.6. Reusability Studies and Removal Efficiency in Artificial Wastewater Sample

EDTA solution (50 mM) was used as a desorption agent for the desorption of Cd(II) ions adsorbed by the composite cryogel cartridges. Prior to the desorption process, the composite cryogel cartridges were washed with deionized water to remove impurities and other unbound residues. Desorption was performed with 100 mL of desorption solution at room temperature for 2 h. After desorption, the cryogels were washed with deionized water. The adsorption–desorption process was performed by reusing the same adsorbent 10 times after regeneration to determine the reusability of composite cryogel cartridges.

The 500 mg/L Cd(II) solution was spiked in 10 mL of the artificial wastewater sample, and then the sample was treated with MIP1 and MIP2 composite cryogel cartridges at a flow rate of 1.0 mL/min at room temperature for 2 h. The diameter and height of the column was 1×5 cm. The initial and final ion concentrations were determined using the inductively coupled plasma mass spectrometry (ICP-MS).

## 3. Results and Discussion

### 3.1. Characterization Studies

The MAC was selected as a functional monomer and HEMA as an assistant monomer. The complexation of chelating monomer MAC and metal ions was performed via metal coordination interactions. The formation of the polymer was also characterized by FTIR-ATR measurements. To investigate the MAC-Cd (II) complex into the polymer structure during the polymerization process, FTIR-ATR spectra of MIP1, MIP2, and NIP composite cryogel cartridges were obtained (Figure 2A). Composite cryogel cartridges have the characteristic stretching vibration bands of O-H at 3368 cm^−1^, C-C at 2944 cm^−1^, and amide I and amide II absorption bands at 2134 and 1992 cm^−1^, respectively. The S-H vibration bands were investigated at 1398 cm^−1^.

The morphological structure of the cryogels was visualized by SEM photographs. The SEM pictures of the interior structure of the cryogels are demonstrated in Figure 2B1,B2. It can be seen that the macroporous structures of the composite cryogel cartridges are opaque and sponge-like. The cryogels were frizzed in at −53 °C in a lyophilizer to avoid the pore loss of the polymers. The structural examination of the MIP composite cryogel cartridges shows us that cryo-porous and non-thick polymer walls have wide and always internally interconnected pores (10–200 μm in diameter). At the same time, the imprinting procedure has increased the capacity. The swelling and porosity properties of MIP1, MIP2, and NIP composite cryogel cartridges are listed in Table 1. 

### 3.2. Optimization of Preparation and Binding Conditions

The effect of the monomer amounts was determined to develop an effective imprinting procedure, and the results are shown in Table 2. Herein, when the Cd(II) amount was fixed at 0.2 mmol, increasing the monomer amount resulted in an increase of the adsorption capacity.

### 3.3. Adsorption of Cd(II) from Aqueous Solutions

The adsorption capacity of Cd(II) ions is highly linked to pH. The rebinding capacity of the Cd(II) ions onto the cryogels was obtained with different pH parameter that varied depending on the nature of the adsorbent and the type of adsorbed ions [44]. The pH of the initiative is one of the significant factors in the adsorption process. The results of the Zeta potential study show that the pH value alters the charge on the surface of the adsorbent, so changes in pH value also alter the adsorption capacity of the adsorbent. The adsorption capacities of Cd(II) onto the MIP1 and MIP2 composite cryogel cartridges were demonstrated in Figure 3A, at different pH values between 4–8. The maximum adsorption capacity of Cd(II) in both of the cryogels cartridges was found at pH 6.4, which can be explained as the cysteine groups of the MAC monomer. Pre-complexation of MAC with Cd(II) was performed in a pH 6.5 phosphate buffer. The isoelectric point of cysteine is 5.02. Therefore, electrostatic interactions are dominant at this pH value. All data were repeated three times.

Figure 3B shows the effect of the concentration of Cd(II) ions on the adsorption capacity at different concentration values between 10–200 mg/L. Adsorption amounts of Cd(II) ions increased rapidly up to a 100 mg/L concentration, but after 100 mg/L, due to filling all of the metal binding area on the surface of the adsorbent the quantities of adsorption were fixed. The adsorption capacity increased with the increasing concentration of Cd(II) ions. As can be seen from the figure, the maximum adsorption capacity of the MIP2 composite cryogel cartridge was higher than that of the MIP1 and NIP composite cryogel cartridges. The maximum capacities obtained for MIP1 and MIP2 were 98.8 and 76.35 μmol/g cryogels. The NIP composite cryogel cartridge did not show a significant amount of adsorption capacity. Adsorption isotherms are used to explain the correlation between the adsorption capacity on the adsorbent and the solute concentration when the two phases are in equilibrium. In this study, Langmuir and Freundlich were used as binding isotherms to explain the relationship between Cd(II) ions and composite cryogel cartridges. The Langmuir model considers monolayer adsorption over the homogeneous adsorbent area. The Freundlich isotherm theory is an exponential equation, whereupon with an increasing adsorbate concentration there is also an increased adsorption on the adsorption surface. Modeling of the equilibrium data was performed using the Langmuir isotherm model and Freundlich isotherm model, given as [45]:Langmuir: *C_e_Q_e_* =1*Q_maxce_* + 1*Q_max_ K_L_*(6)
Freundlich: *Q_e_* = *K_F_* Ce1/*n*(7)
where *Q**_e_* (μmol/g) is the adsorbed amount of Cd(II) ions by the composite cryogel cartridge, *K**_L_* (mL/mg) is the Langmuir isotherm constant, *Q**_max_* (μmol/g) is the maximum adsorption capacity of the composite cryogel cartridge, and *C**_e_* (mg/mL) is the equilibrium Cd(II) concentration. *n* and *K**_F_* are the Freundlich constants related to the adsorption intensity and adsorption capacity of the composite cryogel cartridge, respectively.

Higher correlation coefficient values (R^2^) denote that the experimental data tends to be better fitted with the Langmuir isotherm, which means that the adsorption of Cd(II) onto the composite cryogel cartridge is a monolayer adsorption (Table 3).

The factors affecting the adsorption mechanism, such as mass transfer and the binding itself, were investigated using two different kinetic models. The pseudo-first order and pseudo-second order equations can be used in this case, assuming that the measured concentrations are equal to the adsorbent surface concentration. The first-order rate expression of Lagergren is one of the most widely used for the adsorption of solute from a liquid solution [46]:log(*q_e_* − *q_t_*) = log(*q_e_*) − (*k_1_t*)/2.303(8)
where *q_e_* is the experimental amount of Cd(II) adsorbed at equilibrium (µmol/g), *q_t_* is the amount of Cd(II) adsorbed at time *t* (µmol/g), and *k*_1_ is the rate constant of the pseudo-first order adsorption (min^−1^). A linear dependence of log(*q_e_* − *q_t_*) on *t* suggests the applicability of this kinetic model. Furthermore, in many cases, the pseudo-first order equation of Lagergren does not fit well with the whole range of the contact time and is generally applicable to the initial stage of the adsorption processes. The pseudo-second order kinetic model is expressed by:*t*/*q_t_* = (1/*k*_2_*q*^2^_*eq*_)+(1/*q*_*eq*_)*t*(9)
where *k*_2_ is the rate constant of the pseudo-second order adsorption (mg/L min). If the pseudo-second order kinetics is applicable, the plot of *t/q* versus *t* should be linear. The pseudo-second order kinetic model is favorable when the adsorption behavior over the whole range of adsorption is in agreement with the chemical adsorption being the rate controlling step. The adsorption capacity *q_eq_* obtained in this study was found to be 77.48 and 96.01 µmol/g for the MIP1 and MIP2 composite cryogel cartridges, respectively, with the first-order kinetic model. The adsorption capacity value for Cd(II) obtained from the pseudo-second order adsorption was 69.54 and 87.31 µmol/g, respectively, for the MIP1 and MIP2 cryogels. The order of fitness of these kinetics models to the experimental data was found to be pseudo-first order (R^2^: 0.996) > pseudo- second order (R^2^: 0.896) for the MIP1 cryogel and pseudo- first order (R^2^: 0.982) > pseudo- second order (R^2^: 0.858) for the MIP2 cryogel. The theoretical *q_eq_* value estimated from the pseudo-first order kinetic model was very close to the experimental value, and the correlation coefficient was high. The results indicate that this ion-imprinted adsorbent system was described by the first-order kinetic model.

The effects of the temperature on the adsorption capacity are shown in Figure 3C. The adsorption capacity of the MIP2 composite cryogel cartridge decreased with an increasing temperature from 4 °C to 45 °C. The maximum adsorption capacity was obtained at 4 °C. That result explained the exothermic structure of the adsorption procedure.

### 3.4. Comparison of Adsorption Capacities for the Cd (II) ion of Various Cd (II)-Imprinted Polymers from Aqueous Samples 

In the literature, various ion-imprinted polymers have been reported with a wide range of sorption capacities for Cd(II) ions. They are compared with the results of this study in Table 4. The sorption capacities of the present polymers vary in the range of 10–200 mg/L from aqueous solutions. The table shows a comparison with the maximum sorption capacities of several sorbents found in the literature. MIP1 and MIP2 cryogels have an adsorption uptake of the same order as that of the reported materials. In some of these studies, the adsorption capacity was reported to be greater. However, from a feasibility point of view, the manufacturing of the sorbent involves several steps that are difficult to reproduce on an industrial scale. However, MIP1 and MIP2 cryogels are low-cost materials. Cryogels can be easily developed in an industrial pilot plant.

### 3.5. Selectivity Experiments

Selectivity is a vital and important parameter in molecular imprinting polymers. Therefore, to measure the selectivity of MIP1 and MIP2 composite cryogel cartridges for Cd(II), the competitive ion adsorption studies of Cd(II), Pb(II), and Zn(II) ions as non-templated competitive ions were investigated. The adsorption capacity of each ion is shown in Table 5. The NIP composite cryogel cartridge was used as control material in the experiment. The selectivity ratio of Cd(II) ions was much higher than for the other comparison ions and selectively adsorbed to the composite cryogel cartridge. It has been determined that selectivity is obtained by active cavities. It should be noted that these ions were chosen as competitive metal ions due to their similar ionic radii, similar ionic properties, and due to them having the same charge as Cd(II). The distribution coefficients and selectivity coefficients indicated a significant increase in Cd(II) adsorption. The selectivity coefficients for the MIPs’ cryogel cartridges demonstrated the fact that the competitive ions did not affect the selective removal of Cd(II) ions in the aqueous samples. These results showed that the relative selectivity coefficients of MIP1 and MIP2 cryogels for Pb(II) and Zn(II) were greater than for the non-imprinted matrix. The adsorption capacities obtained for Pb(II) and Zn(II) were 2.4 and 2.06 for MIP1, respectively. Additionally, the adsorption capacities obtained for Pb(II) and Zn(II) were 2.88 and 2.52 for MIP2, respectively. The high selectivity of the MIP cryogels is due to the well-designed coordination geometry of incorporated MAC molecules and Cd(II) ions in the polymerization.

### 3.6. Reusability Studies and Removal Efficiency from an Environmental Water Sample

The high regeneration efficiency for the composite cryogel cartridge is generally a very important feature for making these adsorbents more suitable for applications. Furthermore, this parameter is a critical concern from an economical viewpoint. 50 mM EDTA was used as a desorption solution for 1 h. High reusability ratios of up to 92% were obtained for the composite cryogel cartridge after 10 adsorption–desorption cycles. Therefore, this composite cryogel cartridge can be used several times as an adsorbent for Cd(II) removal. Repeated adsorption/desorption processes showed that these novel composite cryogel cartridges are very suitable for heavy metal removal.

The removal efficiency of Cd(II) species by MIP1 and MIP2 composite cryogel cartridges from artificial wastewater samples was also examined under the optimum experimental adsorption conditions. The maximum capacities obtained for MIP1 and MIP2 were 7.96 and 15.35 μmol/g cryogels. The removal efficiency of Cd(II) was found to be 98.8% in the presence of the other metal ions in the environmental water samples.

## 4. Conclusions

Ion-imprinted materials have been shown to possess a very high degree of selectivity towards targeted substances. This method is the right choice for producing regular polymers that are less costly. Due to its unique recognition technique, the number of recent studies has increased so as to increase the interaction of functional monomers with binding sites. PHEMA-based composite cryogel cartridges provided chemical and physical stability, as well as biocompatibility. The maximum adsorption capacities for Cd(II) ions obtained for MIP1 and MIP2 were 98.8 and 76.35 μmol/g cryogel. This fast adsorption equilibrium is most probably due to high complexation and geometric affinity between Cd(II) ions and Cd(II) cavities in the matrix structure. The adsorption values increased with an increasing concentration of Cd(II) ions. The relative selectivity coefficient is an indicator that expresses an adsorption affinity of recognition sites to the imprinted Cd(II) ions. The MIP1 and MIP2 cryogels can be used many times without decreasing their adsorption capacities significantly. Here, Cd(II) imprinted composite cryogel cartridges demonstrated high specificity and selectivity in Cd(II) ions removal from aqueous solutions. Repeated adsorption/desorption processes showed that these unique composite cryogel cartridges are very suitable for heavy metal removal.

## Figures and Tables

**Figure 1 polymers-12-01149-f001:**
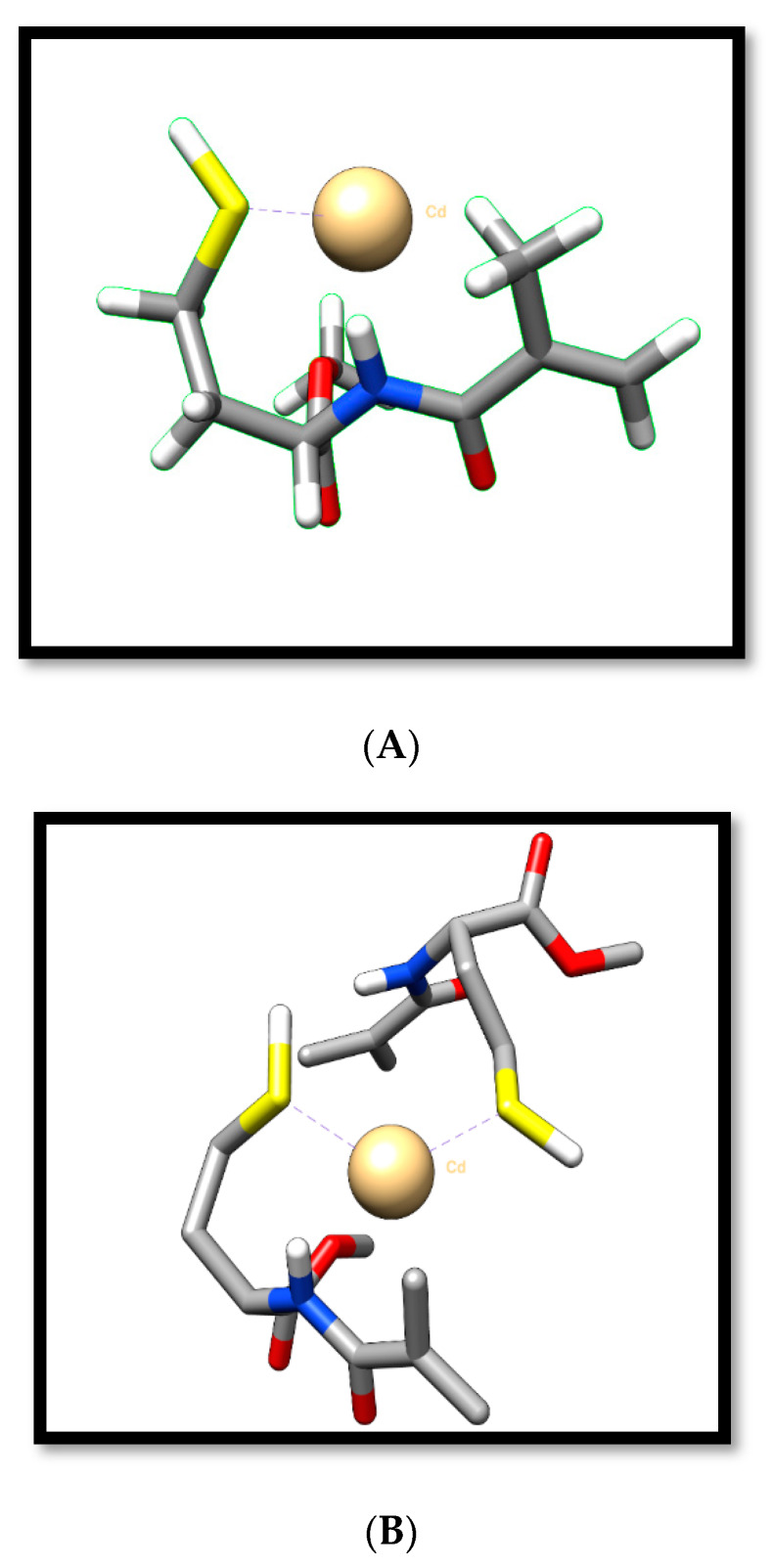
The molecular formulas of the (**A**) 1MAC:1Cd(II) and (**B**) 2MAC:1Cd(II) complex monomer.

**Figure 2 polymers-12-01149-f002:**
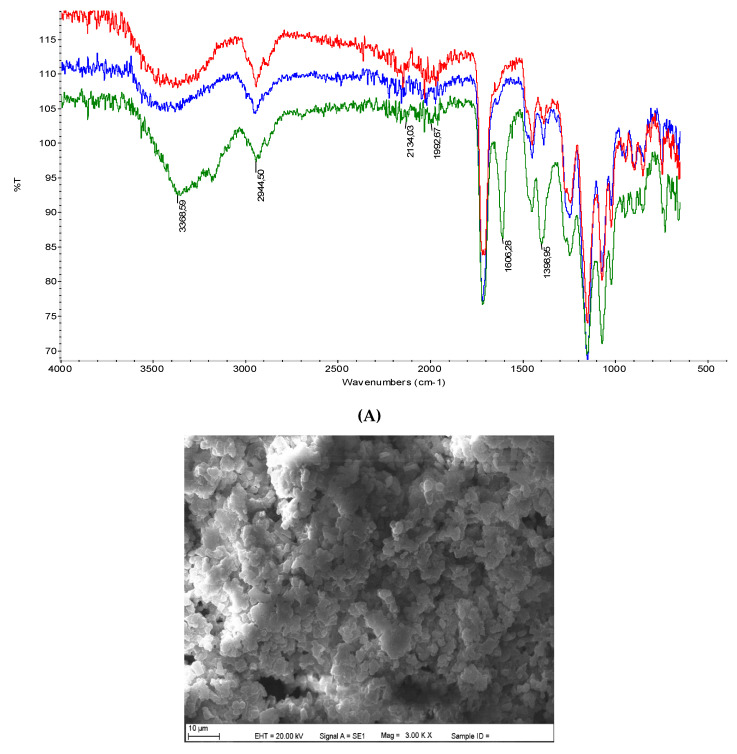
(**A**) FTIR-ATR spectra of MIP2 (red line), MIP1 (blue line), and NIP (green line) composite cryogel cartridges. SEM photographs of (**B1**) MIP1 and (**B2**) MIP2 composite cryogel cartridges.

**Figure 3 polymers-12-01149-f003:**
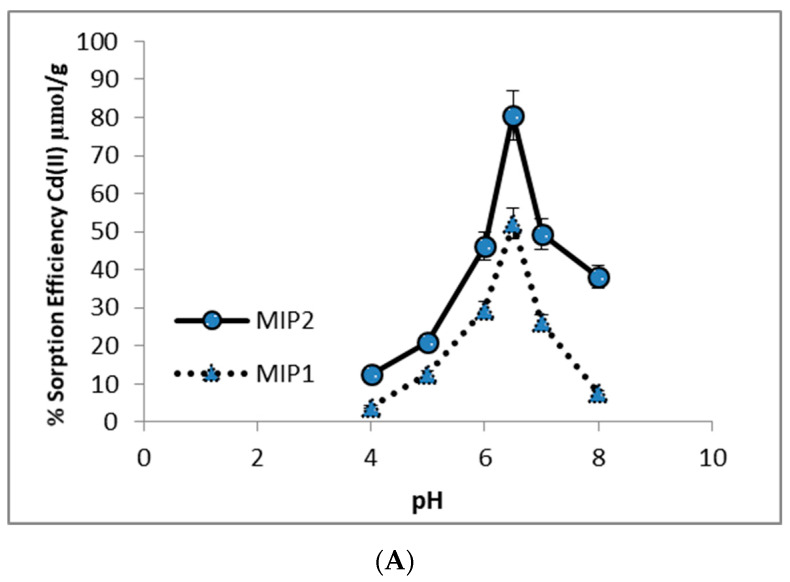
(**A**) Effect of pH on Cd(II) ions adsorption (concentration of Cd(II) 100 mg/L, T: 25 °C, flow rate 0.5 mL/min, time of experiments 120 min). (**B**) Effect of concentration on Cd(II) adsorption (pH 6.4, T: 25 °C, flow rate 0.5 mL/min, time of experiments 120 min). (**C**) Effect of temperature on Cd(II) adsorption (concentration of Cd(II) 75 mg/L, pH 6.4, flow rate 0.5 mL/min, time of experiments 120 min).

**Table 1 polymers-12-01149-t001:** Swelling degree of MIP1, MIP2, and NIP composite cryogel cartridges.

Composite Cryogel Cartridges	Swelling Degree H_2_O g/Polymer
MIP1	8.35 ± 0.12
MIP2	8.72 ± 0.15
NIP	7.58 ± 0.11

**Table 2 polymers-12-01149-t002:** Effect of the monomer ratio on the adsorption capacity.

Polymer	MAC mmol	Cd(II) mmol	Q μmol/g Polymer
MIP1	0.2	0.2	76.35
MIP2	0.4	0.2	98.89
NIP	0.2	-	5.74

**Table 3 polymers-12-01149-t003:** Langmuir and Freundlich adsorption isotherm constants.

Composite Cryogel Cartridge	EXPERIMENTAL	Langmuir			Freundlich		
	Q (μmol/g)	Q_max_ (μmol/g)	b (mL/μmol/g)	R^2^	K_F_	1/n	R^2^
**MIP2**	98.89	97.79	9.2	0.996	44.7	0.851	0.983
**MIP1**	76.35	38.1	1.93	0.992	181.4	0.646	0.901

**Table 4 polymers-12-01149-t004:** Literature comparison of adsorption capacities for cadmium ions.

Monomer	Polymerization Method	Adsorption Capacity	Reference
Allyl thiourea	Surface	0.37 mmol/g	[47]
Graphene Oxide	Surface	0.77 mmol/g	[48]
2-vinylpyridine	Precipitation polymerization	0.14 mmol/g	[49]
β-cyclodextrin	Emulsion	0.95 mmol/g	[50]
Thionine	Surface	0.309 mmol g	[51]
Aminoethyl chitosan	Emulsion	0.23 mmol/g	[52]
p(HEMA-co-MAC)	Emulsion	26.6 µmol/g	[41]
p(HEMA-co-MAC)	Chemisorption	0.05 µmol/g	[53]
p(HEMA-co-MAC)/IIP-Cd	Chemisorption	0.28 µmol/g	[53]
p(PVA-co-HA)	Metal complexation	0.47 mmol/g	[54]
Na-magadiite	hydrothermal medium condition	0.57 mmol/g	[14]
Na-magadiite-Cyanex 272	hydrothermal medium condition	0.44 mmol/g	[14]
MIP2*	Emulsion	98.33 µmol/g	This work

* poly(hydroxyethyl methacrylate N-methacryloly-(L)-cysteine methylester).

**Table 5 polymers-12-01149-t005:** K_d_, k, and k′ proportions of Pb(II), Zn(II), and Cd(II) for MIP1 and MIP2 composite cryogel cartridges.

	NIP		MIP1			MIP2		
Metal ions	K_d_	k	K_d_	k	K’	K_d_	k	K’
Cd(II)	293.6	-	156.6	-	-	377.7	-	-
Pb(II)	240.5	1.22	7.25	21.59	17.68	9.61	39.27	32.16
Zn(II)	253.1	1.16	5.22	29.67	25.58	8.23	45.86	39.53

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
