# Peer review of "Composite Polymeric Cryogel Cartridges for Selective Removal of Cadmium Ions from Aqueous Solutions"

_polymers, 2020, doi:10.3390/polym12051149_

Round 1
Reviewer 1 Report
Manuscript Number: polymers-772185
Dear editor,
In the present paper, composite polymeric cryogel cartridges were studied for removal of cadmium from aqueous solutions. The results presented by the authors could be of interest for wastewater treatment applications. Nevertheless, the manuscript requires major corrections before publication:
- Please do not use abbreviations in the abstract.
- The introduction needs to be improved. It is not clear the relevance of the studied process; the contribution to scientific community is dispersed. Why adsorption technique is important against other processes? I recommend the following recent works for increasing the importance of the introduction:
- Journal of Environmental Chemical Engineering 6(4) (2018), 5351-5360
- Polymers2019, 11(2), 340; https://doi.org/10.3390/polym11020340
- Polymers2019, 11(9), 1509; https://doi.org/10.3390/polym11091509
- It is not clear why a cryogel step of the material preparation is important? What is the influence of this step on the material surface?
- The results of sorption capacity obtained by the authors should be compared with the literature? Please provide a summary table for comparing the sorption capacities with those reported by the literature. Please indicate the sorption capacity in mmol/g
- What is the sorption mechanism of the sorbent?
- Why this polymeric material should be used instead of those commercial ones?
- It is not clear why the sorbent is more selective towards Cd ions? Please in table 4 indicate the sorption capacity of zinc and lead, and compare with cadmium (mmol/g).
- Conclusions should be improved.
Author Response
Manuscript Number: Polymers-772185
Title: " Composite Polymeric Cryogel Cartridges for Selective Removal of Cadmium Ions from Aqueous Solutions"
Polymers
RESPONSE TO REVIEWERS
Thank you very much for reviews. We have considered the comments of Referees’ and made some changes in the manuscript. Our changes and responses are presented in the following. All modification has been marked red colour in the main manuscript file.
Reviewer: 1
- Comments:
In the present paper, composite polymeric cryogel cartridges were studied for removal of cadmium from aqueous solutions. The results presented by the authors could be of interest for wastewater treatment applications. Nevertheless, the manuscript requires major corrections before publication:
Please do not use abbreviations in the abstract.
Answer: Thank you for your careful revision. We revised the abstract part. The changes have been marked red.
The introduction needs to be improved. It is not clear the relevance of the studied process; the contribution to scientific community is dispersed. Why adsorption technique is important against other processes? I recommend the following recent works for increasing the importance of the introduction:
Journal of Environmental Chemical Engineering 6(4) (2018), 5351-5360
Polymers2019, 11(2), 340; https://doi.org/10.3390/polym11020340
Polymers2019, 11(9), 1509; https://doi.org/10.3390/polym11091509
Answer: Thank you for your suggestion. The advised references are added, the introduction has reviewed and goes quite well with the manuscript (page 3, line 1).
It is not clear why a cryogel step of the material preparation is important? What is the influence of this step on the material surface?
The porous surface of cryogel make it efficient for sorption and create cavities for selective attachment on the surface.
The results of sorption capacity obtained by the authors should be compared with the literature? Please provide a summary table for comparing the sorption capacities with those reported by the literature. Please indicate the sorption capacity in mmol/g
Answer: Thank you for your suggestion. We compared the sorption capacity of our study with the literature.
What is the sorption mechanism of the sorbent?
Answer: Thank you. We calculated and wrote in the manuscript as you suggested.
Why this polymeric material should be used instead of those commercial ones?
Answer: Cryogel matrices are cheap and characterized by high sorption capacity towards toxic metal ions. Cryogels provide potential advantages in terms of their low-pressure drop and lack of diffusion resistance as compared to the traditional gel-bead columns. The use of cryogels is promising in cases where a selective removal of metal ions from a complex mixture like wastewater is crucial.
It is not clear why the sorbent is more selective towards Cd ions? Please in table 4 indicate the sorption capacity of zinc and lead, and compare with cadmium (mmol/g).
Answer: Thank you for your suggestion. Table 4 revised and corrected. The sorption capacity of zinc and lead are added in the manuscript (page 16).
Conclusions should be improved.
Answer: Thank you for your valuable time reviewing this paper. We improved the conclusion.
Reviewer 2 Report
The reviewed manuscript “Composite Polymeric Cryogel Cartridges for Selective Removal of Cadmium Ions from Aqueous Solutions” describes the synthesis on selective polymeric sorbents and investigation of their adsorption properties towards Cd (II) ions. The manuscript includes quite interesting and useful experimental results.
Main comments:
- The choice of high toxic Pb (II) and Zn (II) ions for study of selectivity properties of obtained sorbents is not clear. The presence of non-toxic metal ions (Na+, K+, Ca2+, Mg2+ etc.) in real waste or drinking water have more competitive and negative effect on sorption efficiency due to higher concentration. Also, the high affinity and low residual concentration of Cd (II) ions are more important parameters for real application sorption materials. Really, the residual concentration for MPI sorbents is very high and much more than Maximum Contaminant Level (MCL) of 0.005 (mg/L) for Cd (II) in drinking water.
- The advantages of prepared sorbents have supported by providing a comparison data for sorption capacity towards Cd (II) ions. Modified natural and metal oxides sorbents are also cheap and characterized by high sorption capacity towards toxic metal ions (e.g. doi.org/10.1016/j.jwpe.2016.01.005; doi.org/10.1016/j.conbuildmat.2019.03.075; doi.org/10.1016/j.jece.2017.03.041; doi.org/10.1080/10643389.2019.1607442).
- Conclusion section has to rewrite. It should be consist the main results obtained in manuscript with underlining of their novelty.
Minor comments:
- The last paragraph of the introduction section has to revised, its sounds like Abstract. Please, clarify the aim of work, their novelty and main tasks.
- Please, provide full experimental conditions to: metal ions concentration in section 2.5; diameter and height of column in section 2.6; for results in Table 2; time of dynamic test for Fig. 3
- There are no porosity properties in Table 1, please revise it.
- Table 4 have to restructure due to data for NIP is duplicated.
Author Response
Reviewer:2
Comments and Suggestions for Authors
The reviewed manuscript “Composite Polymeric Cryogel Cartridges for Selective Removal of Cadmium Ions from Aqueous Solutions” describes the synthesis on selective polymeric sorbents and investigation of their adsorption properties towards Cd (II) ions. The manuscript includes quite interesting and useful experimental results.
Main comments:
The choice of high toxic Pb (II) and Zn (II) ions for study of selectivity properties of obtained sorbents is not clear. The presence of non-toxic metal ions (Na+, K+, Ca2+, Mg2+ etc.) in real waste or drinking water have more competitive and negative effect on sorption efficiency due to higher concentration. Also, the high affinity and low residual concentration of Cd (II) ions are more important parameters for real application sorption materials. Really, the residual concentration for MPI sorbents is very high and much more than Maximum Contaminant Level (MCL) of 0.005 (mg/L) for Cd (II) in drinking water.
The advantages of prepared sorbents have supported by providing a comparison data for sorption capacity towards Cd (II) ions. Modified natural and metal oxides sorbents are also cheap and characterized by high sorption capacity towards toxic metal ions (e.g. doi.org/10.1016/j.jwpe.2016.01.005; doi.org/10.1016/j.conbuildmat.2019.03.075; doi.org/10.1016/j.jece.2017.03.041; doi.org/10.1080/10643389.2019.1607442).
Conclusion section has to rewrite. It should be consist of the main results obtained in manuscript with underlining of their novelty.
Answer: Thank you for your valuable time reviewing this paper, and your suggestion. We added these references.
Minor comments:
The last paragraph of the introduction section has to revised, its sounds like Abstract. Please, clarify the aim of work, their novelty and main tasks.
Answer: Thank you for your careful revision. We revised the abstract part. The changes have been marked red.
Please, provide full experimental conditions to: metal ions concentration in section 2.5; diameter and height of column in section 2.6; for results in Table 2; time of dynamic test for Fig. 3
There are no porosity properties in Table 1, please revise it.
Table 4 have to restructure due to data for NIP is duplicated.
Answer: Thank you for your careful revision. We revised all of them as you suggested.
Round 2
Reviewer 1 Report
Dear Editor,
The manuscript was improved by the authors. However, some corrections are still required before publication.
- In section 3.4 (Table 4), the authors have provided the sorption capacity of the sorbents and compared with the values reported by the literature, but is not clear why the studied MIP1 and MIP2 sorbents are advantageous in comparison with other sorbents. The sorption capacity obtained by the authors is very limited (lower than other sorbents). Please improve the discussion in this section.
In Table 4, please use the same unity of sorption capacity (q) mmol/g or µmol/g (i.e., all sorbents with the same unity). I recommend including the values obtained by Attar et al., 2018 (the reference [14]) to this table.
Author Response
Manuscript Number: Polymers-772185-R2
Title: " Composite Polymeric Cryogel Cartridges for Selective Removal of Cadmium Ions from Aqueous Solutions"
Polymers
Reviewer: 1
- Comments:
Dear Editor,
The manuscript was improved by the authors. However, some corrections are still required before publication.
- In section 3.4 (Table 4), the authors have provided the sorption capacity of the sorbents and compared with the values reported by the literature, but is not clear why the studied MIP1 and MIP2 sorbents are advantageous in comparison with other sorbents. The sorption capacity obtained by the authors is very limited (lower than other sorbents). Please improve the discussion in this section.
In Table 4, please use the same unity of sorption capacity (q) mmol/g or µmol/g (i.e., all sorbents with the same unity). I recommend including the values obtained by Attar et al., 2018 (the reference [14]) to this table.
Answer: Thank you for your revision. We revised Table 4. The changes have been marked red. The table shows a comparison with the maximum sorption capacities of several sorbents found in the literature. In some of these studies, the adsorption capacity was obtained greater. But, from a feasibility point of view, the manufacturing of the sorbent involves several steps which are difficult to reproduce on an industrial scale. However, these prepared cryogels are low-cost material. Cryogels easily can be developed in an industrial pilot plant.
Reviewer 2 Report
The revised manuscript could be accepted.
Author Response
Answer: Thank you for your revision.